# Human-like individual differences emerge from random weight initializations in neural networks

**Herrick Fung, N. Apurva Ratan Murty, Dobromir Rahnev**
School of Psychology
Georgia Institute of Technology
Atlanta, GA 30332
`herrickfung@gmail.com, ratan@gatech.edu, rahnev@psych.gatech.edu`

## Abstract

Much of AI research targets the behavior of an *average* human, a focus that traces to Turing's imitation game. Yet, no two human individuals behave exactly alike. In this study, we show that artificial neural networks (ANNs) trained with different random initializations exhibit substantial individual differences that resemble those in humans. Using a large dataset (N = 60) of human responses (accuracy, confidence, & response time) in a digit recognition task, we trained multiple instances of three ANN architectures on the same task, creating as many ANN instances as human subjects. We found that these ANN instances vary significantly from one another. Critically, ANN instances showed consistent variation in their alignment with specific human subjects. This consistency in alignment between ANN instances and humans extended across behavioral metrics, indicating that an ANN instance mimicking an individual on one metric also does so on others. Finally, we showed that leveraging these alignments improves predictions of individual human responses. Our findings highlight the potential of ANNs to capture human variability, opening new directions to develop models that go beyond aligning the *average* human and instead aligning the idiosyncratic behavior of specific *individuals*.

## 1  Introduction

Artificial neural networks (ANNs) have become the leading computational model of the primate visual system [1–7]. Deep convolutional architectures trained on large datasets achieve human-level performance [8, 9], demonstrate similar error patterns [10], and exhibit signatures of human perceptual decision making [11] in complex image recognition tasks. Since Turing's original proposal of the imitation game [12], much of the AI (and now NeuroAI [13]) research has been evaluated against the behavior of an *average* human, overlooking a fundamental fact: individual humans perceive, think, and behave differently. Given ANNs' success at perceptual tasks, we ask: do ANNs with the same architecture also exhibit individual differences in behavior? If so, do these individual differences in ANNs capture the range of variability observed across human individuals? Can these individual differences in ANNs serve as computational proxy models for individual differences in human behavior? Here, we show that neural networks trained with different initializations exhibit substantial individual differences that resemble those observed in humans.

Prior work has demonstrated that seemingly minor factors such as weight initialization [14] or the order of training images [15] can drive substantial variability in the features a model learns. Even though the variability in models' internal representations has been investigated, far less attention has been paid to whether different model instances produce distinct behavioral patterns. Specifically, no work to date has investigated how such individual differences in ANNs relate to the well-documented

individual differences in humans. Thus, it remains an open question whether the idiosyncracies of ANNs capture the spectrum of human behavioral variability.

We make three main contributions. We demonstrate that (1) ANN instances differing only in their random initialization exhibit significant yet consistent variation in alignment with specific human subjects, paralleling the variability and stability in human-human alignment (Figure 1), (2) human-ANN alignment generalizes across behavioral metrics: an ANN instance best matches an individual on one metric also tends to mimic the same individual on other metrics (Figure 2), (3) leveraging these human-ANN alignments improves predictions of individual behavioral responses in held-out data (Figure 3).

## 2 Methods

### 2.1 Human data

We reanalyzed publicly available data from an existing study [11]. In the original experiment, 60 human subjects performed an 8-choice noisy digit recognition task. The stimulus set consisted of a total of 480 unique images randomly selected from the MNIST validation dataset [16]. Each image was tested twice, producing a total of 960 trials. Each trial began with subject fixating on a white fixation cross for 500 to 1000 ms, followed by an image of a handwritten digit infused with two levels of random uniform noise, shown for 300 ms. Subjects then sequentially reported their choice and confidence (4-point rating scale) with no time constraints. The data contained accuracy, confidence, and response time (RT) on each trial.

### 2.2 Neural network data

We trained ANNs of three different architectures—RTNet [11], AlexNet [8], and ResNet18 [9]. For each architecture, we trained 60 unique instances (matching the number of human subjects) by changing only the random weight initialization. All models were trained on the MNIST training dataset to reach at least 97% accuracy on the validation dataset. Models were then tested on the same 480 images that were presented to the human subjects in the experiment. We matched the accuracy of all models at the group level by adjusting the noise level in the image. Confidence for all model architectures was defined as the logit margin between the first and second highest predicted classes, a method shown in a recent work to better predict human confidence than alternative approaches [17]. Below, we detail the specifics for each architecture.

**RTNet:** RTNet is a recently developed neural network designed to capture key features of human perceptual decision-making and predict RT [11]. Unlike standard feedforward CNNs, RTNet is a Bayesian neural network with probabilistic weights and incorporates an evidence accumulation mechanism, repeatedly processing an image until the accumulated evidence for a choice reaches a threshold. RT can thus be defined by the number of repetitions needed to reach this threshold. We used the published dataset of 60 instances of the RTNet that differ only in their random initialization during training [11]. These instances were trained for 15 epochs with a batch size of 500, using the ELBO loss function and Adam optimizer with the default parameters.

**AlexNet & ResNet18:** 60 instances of the AlexNet [8] and 60 instances of the ResNet18 [9] were trained for 15 epochs with a batch size of 128, using the cross-entropy loss function and the Adam optimizer with the default parameters.

### 2.3 Analytic framework

To assess alignment between ANN instances and human subjects, we computed pairwise correlations between all human subjects' and ANN instances' responses to obtain a $60 \times 60$ similarity matrix for each model architecture, where higher values indicate better alignment between a human subject and an ANN instance. This procedure was repeated over 1000 bootstrap iterations with random image splits to estimate consistency (Figure 1c & 2b), calculated as the within-subject correlation across splits. We then corrected this correlation by subtracting the average across-subject correlation to account for similarity expected by shared correlations. To test whether these alignments could improve behavioral predictions for unseen data, we computed the similarity matrix from one subset to estimate the human-ANN alignments, which then served as weights to predict behavior in the held-

out subset (Figure 3). We then compared this alignment-weighted prediction to an equal-weighted average prediction to assess its improvement. For benchmarking, a human-human counterpart of these analyses was performed by replacing 60 ANN instances' responses with all human subjects except the target (60 - 1), yielding a $60 \times 59$ similarity matrix that measures the alignment between the target subject and all other individual subjects except themselves.

## 3   Results

### 3.1   ANN instances show differences in alignment with specific human subjects

We first established that individual differences arise in both ANN instances and human subjects (Figure 1a). Although these ANN models were trained on noiseless images and evaluated on images with added noise and differed only in their random initialization seeds, these ANN models demonstrated a large range of accuracy ranging from 21% to 93%, with all ANN architectures demonstrating greater variability than the variability observed in human subjects (41% - 81%). Similar results were found for confidence but not RT, where we found larger variability in humans than RTNet.

Having established individual differences in both ANNs and humans, we next explored whether individual ANN instances vary in how well they align with specific human subjects. Across all three ANN architectures, we observed substantial variability in the human-ANN similarity matrices, most of which was statistically indistinguishable from the human-human variability (all $p's > 0.257$; Figure 1b). The only exception was ResNet18 which demonstrated reliably lower variability (both $p's < 0.011$). To verify that the observed variability in human-ANN alignment was not spurious but instead reflected meaningful variability among ANN instances, we assessed human-ANN alignment consistency using separate sets of images (Figure 1c). Across 1000 bootstrap samples, human-RTNet and human-AlexNet alignments were as consistent as the human-human benchmark for all behavioral metrics (all $p's > 0.19$). In contrast, ResNet18 showed lower consistency than the human-human benchmark for both accuracy and confidence (both $p's = 0.002$). These results demonstrate that individual ANN instances of RTNet and AlexNet align reliably with specific human subjects, exhibiting variability and consistency comparable to that observed among humans themselves.

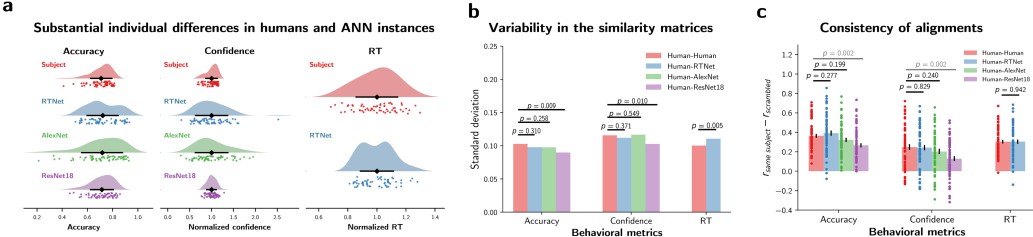

Figure 1: **ANN instances vary in alignment with specific human subjects in a consistent manner. a.** Substantial individual differences in humans and ANN instances. Confidence and RT are normalized by their mean across subjects or instances for visualization. **b.** Variability in aligning human to human and human to ANN instances. P-values: Pairwise differences tested with 2000 bootstraps. **c.** Consistency of human-human and human-ANN alignments between data subsets. Error bars: SEM. Dots: Individual human subjects. P-values: Significance of paired t-tests.

### 3.2   ANN instances that mimic an individual on one metric also mimic the same individual on other metrics

Having shown that ANN instances align with human subjects on each behavioral metric (accuracy, confidence, RT) individually, we next examined whether the alignment between ANN instances and humans generalizes across metrics. Across all pairs of behavioral metrics, human-ANN alignments were significantly positive (all $p's < 0.001$, Figure 2a), indicating systematic correspondence between humans and ANNs even across behavioral metrics. Moreover, except for the human-ResNet18 correlation, which was significantly worse than the human-human benchmark ($p = 2.87 \times 10^{-13}$), all other human-ANN alignments across behavioral metric pairs were indistinguishable from the

human-human benchmark (each $p > 0.18$). Notably, for the case of human-RTNet and RT-confidence, it even outperformed the human-human benchmark ($p = 1.30 \times 10^{-9}$).

To ensure robustness, we validated the across-metric consistency using 1000 bootstrap samples (Figure 2b). We assessed the across-metric consistency by comparing split 1 of one metric to split 2 of another, ruling out the possibility that the observed consistency was driven by shared data variance. Human-RTNet and Human-AlexNet alignments again showed across-metric consistency comparable to the human benchmark in all metric pairs (all $p's > 0.05$), whereas human-ResNet18 alignment was significantly weaker than the human benchmark ($p = 2.72 \times 10^{-6}$). These findings indicate that the correspondence between individual differences in humans and ANN instances extends across multiple behavioral metrics, supporting that ANN instances may capture individuals' diverse behavioral characteristics at a more fundamental level.

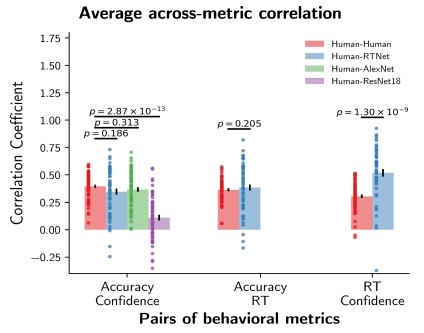 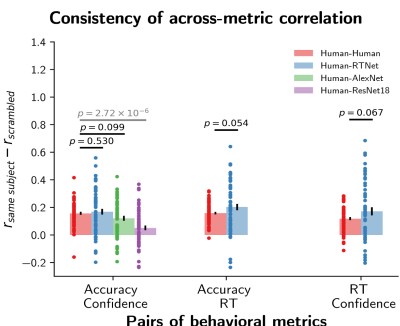

Figure 2: **ANN instances that mimic an individual on one metric also mimic the same individual on other metrics. a.** Average across-metric correlation. **b.** Across-metric consistency of human-human and human-ANN alignments. Error bars: SEM. Dots: Individual human subjects. P-values: Significance of paired t-tests.

### 3.3 Leveraging individual differences in ANN improves single-subject response prediction

So far, we demonstrated that neural networks trained with different initialization exhibit substantial individual differences that align consistently with individual human subjects across both data subsets and behavioral metrics. We next asked if these human-ANN alignments can be leveraged for practical applications. Specifically, we tested whether this alignment could improve predictions of individual human behavior in held-out data. We first evaluated the within-metric prediction and found that incorporating these human-ANN alignments improved predictions for unseen data across all behavioral metrics (all $p's < 0.001$; Figure 3a). We further compared these improvements with the subject counterparts and found that all three ANNs either showed comparable improvements or improve significantly more than the subject benchmark (Figure 3a). Similar benefits were also observed in most across-metric predictions, though the effect was generally weaker as compared to the within-metric predictions (Figure 3b) but are expected because of cross-validation. These results demonstrate that leveraging individual differences in ANN instances improves single-subject response prediction for both within-metric and across-metric contexts.

## 4   Discussion

We investigated the extent to which ANN instances that only differ in their random initializations exhibit individual differences and whether individual differences in ANN instances mimic individual differences in humans. We found that ANN instances demonstrated substantial behavioral variability. Critically, these individual differences closely resembled those observed in humans: the alignment between individual human subjects and individual ANN instances was often comparable to, and in some cases exceeded, the consistency among humans themselves (Figure 1). These alignment consistencies held across different data subsets and even behavioral metrics (Figure 2), suggesting that ANN instances can capture individual human behavior at a deeper level. Finally, we demonstrated that these human-ANN alignments could be leveraged to improve predictions of individual responses in held-out data (Figure 3).

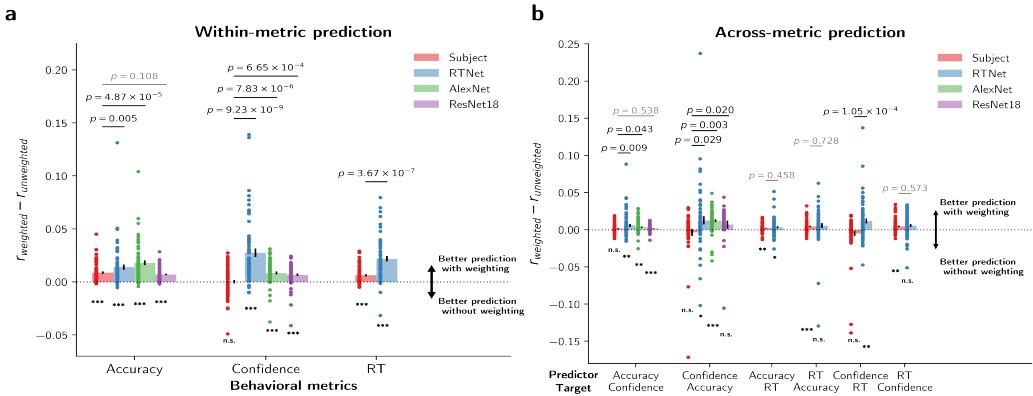

Figure 3: **Leveraging individual differences in ANN instances to improve single-subject response prediction. a.** Within-metric prediction. **b.** Across-metric prediction. Error bars: SEM. Dots: Individual human subjects. P-values: Significance of paired t-tests. Asterisks: One-sample t-test vs. zero *n.s.*: $p > 0.05$; * $p < 0.05$; ** $p < 0.01$; *** $p < 0.001$.

Our findings connect to a broader trend in AI research, which has shifted from merely achieving human-level performance toward crafting models that behave more like humans. For instance, aligning the visual diet of neural networks with the developmental trajectory of the human visual system has been shown to reduce texture bias and improve robustness to different forms of visual noise [18–21], thereby bringing model perception closer to human perception. Efforts in this space have primarily evaluated success using behavioral similarity [11, 12, 18, 20], behavioral and neural predictivity (e.g. BrainScore [22]), and representational similarity [1, 23–25]. Our work introduces a complementary and largely overlooked dimension by examining the alignment between individual variability in ANNs and humans. Specifically, future work should focus not only on mimicking and explaining the average human behavior [12, 13], but also on capturing the structured idiosyncrasies in human behavior. As we show, not all network architectures capture these individual differences in human behavior. ResNet18, for instance, underperforms in mimcking human behavioral variability despite matched accuracy and variance, illustrating that not all variability is equivalent and humans have a distinctive structure to their variability. Future work should also examine how other factors, i.e. training image order, stimulus frequency, category distribution [15], and network architecture can better capture human-like variability.

Taken together, our findings demonstrate that some ANNs trained with different initializations can capture individual differences in human behavior, opening new avenues for developing models that reflect the idiosyncrasy in human behavior.

## Acknowledgments and Disclosure of Funding

The authors declared no competing interests. This work was supported by the National Institute of Health (award: R01MH119189) and the Office of Naval Research (award: N00014-20-1-2622). We thank Alish Dipani and members of the Computation of Subjective Perception Lab and the Visual Cognition Computation Lab for their helpful suggestions and discussions.

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
