# OpenReview forum: "Human-like individual differences emerge from random weight initializations in neural networks"
_NeurIPS.cc/2025/Workshop/UniReps — UniReps2025_

### Official Review · Reviewer_L5QZ · 2025-09-13
**Initialization-Dependent Neural Variability Mirrors Human Individual Differences: A Strong Proof-of-Concept with Methodological Considerations**

**Confidence:** 4

**Review:**

# Summary

This extended abstract explores whether artificial neural networks (ANNs) with different random initializations can develop individual differences similar to human behavior. The authors compared 60 human participants performing a noisy digit recognition task with 60 separately initialized instances of RTNet, AlexNet, and ResNet18. They examined how well each network aligned with specific human subjects in terms of accuracy, confidence, and response time. Results show that ANN instances consistently mirror the variability found between humans: networks that align with a person on one metric also tend to align on others. Moreover, exploiting these alignments improves predictions of individual responses on new data.

# Strengths and Weaknesses

## Strengths

### Novel Conceptual Framework and Methodological Rigor

The author introduces a systematic approach to a fundamentally understudied question in computational modeling of human cognition. The bootstrap validation methodology ($N=1000$) with cross-validation procedures demonstrates appropriate statistical rigor. The human-human baseline provides necessary benchmarking for interpreting ANN-human alignments. The consistency analysis across data subsets effectively validates that observed alignments reflect genuine patterns rather than overfitting artifacts.

### Cross-Metric Generalization Evidence

The finding that ANN instances aligning with specific humans on one metric tend to align on other metrics provides compelling evidence for systematic rather than coincidental correspondence. This cross-metric consistency suggests the networks capture fundamental aspects of individual decision-making patterns rather than task-specific artifacts.

### Practical Applications and Digital Twin Potential

The demonstrated improvement in individual response prediction establishes proof-of-concept for personalized AI applications. Under the framework of computational ``digital twins'', this work opens pathways toward individual-specific neural models for cognitive assessment, personalized interfaces, and precision approaches to human-computer interaction.

## Weaknesses

### Confidence Measurement Equivalence Assumptions

The extended abstract assumes network confidence (logit margin: $\text{confidence} = \log p_1 - \log p_2$) equals human confidence (4-point Likert scale) without adequate validation. While citing evidence that logit margins better predict human confidence than maximum probability, this assumption equates fundamentally different measurements:

\begin{equation}
C_{\text{human}} \in \{1,2,3,4\} \quad \text{vs.} \quad C_{\text{network}} = \log\frac{p_{\max}}{p_{2nd}} \in \mathbb{R}^+
\end{equation}

Human confidence reflects subjective metacognitive processes influenced by uncertainty estimation, prior experience, and calibration biases. Network confidence represents objective mathematical differences between predictions. This methodological assumption could systematically bias observed correlations and requires validation through collection of human second-choice predictions and confidence intervals, which is a closer comparison to the logit margin metric.

### Limited Connection to Neuroscience Literature

The work would benefit from stronger connections to neuroscience literature on individual differences, which could enhance its relevance for the NeurIPS community. Known neural mechanisms underlying human individual differences include:

- Cortical thickness variations correlating with cognitive performance differences
- White matter connectivity patterns affecting processing speed and efficiency
- Neural oscillation patterns varying systematically across individuals
- Neuromodulatory system differences influencing attention and decision-making

Future work could explore whether ANN architectural features correspond to these biological substrates or whether observed patterns reflect computational principles that could inform our understanding of neural computation.

### ResNet18 Performance Analysis and Architectural Understanding

The authors mention ResNet18's limited performance but provide insufficient mechanistic explanation. ResNet18's residual connections create identity mappings:

\begin{equation}
\mathbf{x}_{l+1} = \mathbf{x}_l + \mathcal{F}(\mathbf{x}_l, W_l)
\end{equation}

These residual pathways preserve information across layers, potentially reducing sensitivity to initialization variability. The stability-variability trade-off may explain why ResNet18 shows less individual difference expression: the architecture's design principle of stable gradient flow could conflict with the initialization-dependent variability needed for individual difference expression.

### Limited Task Scope and Generalization Questions

Evaluation focuses on MNIST digit recognition, a relatively simple perceptual task that may not fully capture the breadth of human cognitive variability. Individual differences often emerge more prominently in complex reasoning, ambiguous decision-making, and multi-step problem-solving. The authors acknowledge this limitation, but extending to more complex tasks would strengthen claims about human behavioral modeling.

### Human Factor Analysis

The absence of participant characteristic analysis represents a methodological limitation. Individual differences in cognitive abilities (working memory capacity, processing speed), personality traits (conscientiousness, openness), demographic factors (age, education), and state variables (fatigue, attention) likely influence both task performance and ANN alignment patterns. Controlling for these factors could clarify whether observed correlations reflect computational correspondence or confounded variables.

### Proposed Metric

To formalize the alignment phenomenon, I would propose an Individual Difference Alignment Metric (IDAM) that captures the systematic correspondence between human and network behavioral patterns:

$IDAM_{h,n} = \sum_m w_m \cdot ( \rho(H_{h,m}, N_{n,m}) - \bar\rho_{m} ) / \sigma_{\rho,m} $



where $\rho(H_{h,m}, N_{n,m})$ represents the correlation between human $h$ and network instance $n$ on metric $m \in \{acc, conf, rt \}$; $\bar{\rho}_m$ and $\sigma \{\rho_m}$ are the mean and standard deviation of correlations for metric $m$, and $w_m$ are metric-specific weights. This formulation allows testing whether:

$$\max_n IDAM_{h,n} > IDAM_{baseline}$$

indicating systematic rather than random alignment patterns.

# Quality: 3 (Good)

The experimental methodology demonstrates technical soundness with appropriate statistical controls and cross-validation procedures. Claims are generally well-supported within the restricted scope, though confidence measurement assumptions and limited task complexity constrain broader generalizations. The authors provide reasonable methodological detail for replication but could be more explicit about generalizability limitations of the results.

# Clarity: 2 (Fair)

The presentation suffers from technical density that could obscure core insights for some readers. The methodology requires careful reading to understand fully, and statistical procedures sometimes overshadow conceptual development. The paper would benefit from concrete examples illustrating human-network alignments and clearer presentation of the experimental structure through diagrams or pipelines. Missing specific participant-machine alikeness analysis further limits interpretability. It focuses on reproducibility and documentation of the experiment, rather on insights, discovered patterns or relevant results.

# Significance: 3 (Good)

For personalized AI and computational modeling of individual differences, this represents valuable work establishing proof-of-concept for ``digital twin'' applications. The single-task limitation and limited biological connections constrain broader immediate impact, but the cross-metric consistency findings suggest fundamental rather than superficial phenomena. Extension to complex tasks would strengthen practical relevance and could significantly enhance the contribution's impact.

# Originality: 3 (Good)

The systematic connection between initialization-dependent ANN variability and human individual differences provides novel insights into computational modeling of cognition. While building on established techniques, the specific application represents meaningful innovation. The cross-metric consistency analysis offers original evidence for structured rather than random correspondence patterns.

# Questions

## 1. Neuroscience Connections and Neural Architecture Mapping

How might the observed ANN individual differences relate to known neuroscience of human variability? Given that RTNet's evidence accumulation mechanism resembles parietal cortex drift-diffusion processes while AlexNet's convolutions parallel ventral stream hierarchical processing, do architectural features that better correspond to biological substrates show stronger human alignment? Could this framework help identify which computational operations most closely match specific neural mechanisms?

_Evaluation Impact_: Connecting findings to neuroscience literature could absolutely boost significance in and outside NeurIPS audience.

## 2. Confidence Measurement Validation and Alternative Approaches

The logit margin assumption would benefit from validation. How would results change using alternative confidence measures (entropy-based uncertainty, ensemble disagreement, temperature-scaled probabilities)? Could the authors collect human confidence intervals and second-choice predictions to better parallel network computations?

_Evaluation Impact_: Addressing confidence measurement validity could strengthen methodological foundations and increase quality rating.

## 3. Human Individual Differences Integration

What participant characteristics might predict ANN alignment patterns? Do individuals with higher working memory capacity align better with specific network architectures? How might personality traits (conscientiousness, cognitive flexibility) correlate with network alignment strength?

_Evaluation Impact_: Systematic human factors analysis could increase both quality and significance ratings.

## 4. Task Complexity and Generalization Potential

How might the alignment phenomenon extend to more complex reasoning tasks requiring multi-step problem-solving, ambiguous instruction interpretation, or creative generation? Do initialization-dependent patterns persist in tasks demanding higher-order cognitive processes?

_Evaluation Impact_: Demonstrating generalization beyond simple perceptual tasks would strengthen practical relevance and significance.

# Limitations

### Partially addressed.
The authors could more comprehensively address several important limitations:

- **Task Scope Constraints:** More explicit acknowledgment that MNIST digit recognition, while valuable for proof-of-concept, may not fully capture human cognitive variability
- **Confidence Measurement Validity:** Recognition of fundamental differences between subjective human confidence and objective network margins, with discussion of validation approaches
- **Human Factors Consideration:** Discussion of participant characteristics that could influence observed alignments
- **Architectural Specificity:** More detailed analysis of why certain architectures succeed while others show limited individual difference expression

# Recommendations:
Consider adding a more comprehensive limitations section addressing task generalization constraints, confidence measurement assumptions, and potential biological correspondences. Include discussion of participant characteristic effects and architectural design principles affecting individual difference expression. While the work provides valuable proof-of-concept for personalized computational modeling, broader claims about human behavioral replication would benefit from additional validation across more complex tasks and diverse populations.

**Score:**

3

**Topic Fit:**

3

---

### Official Review · Reviewer_GM6a · 2025-09-15
**Human-like individual differences emerge from random weight initializations in neural networks**

**Confidence:** 3

**Review:**

Summary:
In this paper, the authors investigate how different initializations of artificial neural networks (ANNs) can mimic the variability observed in the outputs of a classification task in humans. They use three ANN architectures, ResNet18, AlexNet, and RTNet, a recently developed Bayesian neural network.  To compare the variance in the outputs of the ANN to human performance, they study the correlations between the responses from the human dataset and the ANN instances. The authors find that RTNet and AlexNet produce response distributions similar to those of the human dataset. The authors propose that ANNs with different initializations could capture the idiosyncrasies seen in human behavior.

Strengths:
The work is novel in that it demonstrates, in particular, RTNet's ability to reproduce the response time variation observed in human data, serving as a model of individual human responses rather than average behavior.
This work supports the digital twin line of work that’s ongoing in the field, even though the ANNs capture only one task.
The methods were rigorous, but could be clarified much better.
The paper studies alignment in the distributions of responses, rather than the responses themselves, which is a novel approach, as far as I know.

Weaknesses:
What is referred to as alignment is somewhat unclear, and I had to make some assumptions.
It is unclear how pairwise correlations were calculated to quantify the alignment.
The consistency metric was hard to understand.
Some interesting observations were not noted in the paper, for example, why the RTNet shows a bimodal distribution in the RT distribution (Figure 1a).

**Score:**

4

**Topic Fit:**

2

---

### Official Review · Reviewer_J9kd · 2025-09-15
**Differences in ANNs due to different random weight initializations resemble individual differences in humans. Paper introduces novelty**

**Confidence:** 3

**Review:**

The paper at hand investigates how different random initializations of ANN yield individual differences that can be compared to humans. To that end, three different ANN architectures (RTNet, AlexNet, and ResNet18) were trained on the MNIST dataset. For each architecture, 60 instances were trained, then tested on 480 images, the same images that were used in a referenced experiment where 60 humans performed a noisy digit recognition task. Quantitative results show individual differences in the ANN instances of RTNet and AlexNet that are in alignment with human subjects. When it comes to alignment across multiple metrics, the alignment between ANN instances and human subjects is apparent for one ANN (RTNet), The results for Human-AlexNet on all pairs was not reported and reported to be weak for human-ResNet18. Reported results show improved single-subject response performance on unseen data when incorporating individual differences in ANN.

**Strengths**
- The paper introduces novelty by showing how random initializations of ANN yield variations that align with specific humans subjects.
- The paper conducts extensive experiments that support their findings.
- The paper shows that the differences in ANN from the random initialization can be leveraged for a better single-subject response prediction.

**Minor weaknesses**
- Unclear statement “ We assessed the across-metric consistency by comparing split 1 of one metric to split 2 of another”.
- The paper states that Human-RTNet and Human-AlexNet alignments showed across-metric consistency comparable to the human benchmark in all metric pairs, however, Figure 2 shows results only of Human-RTNet on all pairs of metrics.
- Experiment details for 3.3 are hard to follow, it is not completely clear what experimental steps led to the results reported in Figure 3.

**Score:**

4

**Topic Fit:**

3

---

### Official Review · Reviewer_41Qw · 2025-09-15
**Good proof of concept on how to investigate individual differences in ANN-Human comparisons.**

**Confidence:** 4

**Review:**

This paper highlights that behavioral variability from different seeds of a trained model mimics the behavioral variability across humans. This hold for the mnist dataset when the model architectures are AlexNet and RTNet although does not work with ResNet18.

Pros:
- The motivation is clear
- This is timely work, as many are using model-model comparisons with multiple seeds as a testbed for model brain comparisons (i.e., Thobani et al. 2025, Bo et al. 2025)

Concerns:
- Some of the claims seem strong for the results:
    - Fig 1: While (b) suggests that the distributions are similar as the stdev is indistinguishable, you can visually see in (a) that there is differences in the structures of the distributions. Both RTnet and AlexNet seem to have different structures to the humans (especially the differences in variability above and below the center). Visually the similarity is best with ResNet although later on we see that this fails elsewhere. A non-parametric test of the distributions may be more appropriate.
- The axes of correlation coefficients are poorly scaled as they go over 1. They are also not consistent across plots, making it difficult to compare.
- The dataset, mnist is extremely low dimensional, with 2/3 models being extremely overparameterized for the task.
- Does it make sense for the effect to be lowest for subject in Fig 3? How do we interpret that?

Conclusions:
- This work is very timely and a good proof of concept. I would like to see it extended to more complex tasks where the model is not as overparametrized.
- It seems like there are more differences in variability between humans and models that are just not captured by the analysis. Non-parametric tests, or other comparisons, might be interesting. Instead of comparing only accuracy and confidence, comparing confusion matrices, the overall distributions, or other metrics might highlight more differences in these distributions.

**Score:**

3

**Topic Fit:**

2